# Spinal cord lesions in relapsing-remitting multiple sclerosis: Prognostic value of intrathecal production of immunoglobulins

Daniel Kreiter[1,2]*, Niels Franken[3], Demmie Bouweriks[1,2], Stephanie Knippenberg[1,2], Raymond Hupperts[1,2], Oliver Gerlach[1,2]

1 Academic MS Center Zuyd, Department of Neurology, Zuyderland MC, Sittard-Geleen, The Netherlands, 2 School for Mental Health and Neuroscience, Department of Neurology, Maastricht University Medical Center, Maastricht, The Netherlands, 3 Faculty of Health, Medicine & Life Sciences, Maastricht University, Maastricht, The Netherlands

* d.kreiter@zuyderland.nl

## Abstract

### Background

Controversy exists about the radiological follow-up of the spinal cord in MS, since cord lesions are mostly symptomatic, but there are also reports that 10% − 25% are asymptomatic. Therefore, a need exists for biomarkers to identify patients prone to future cord involvement where routine cord imaging at follow-up is warranted. Elevated intrathecal IgM has been shown to be cross-sectionally associated with more cord lesions. This study investigated whether there also is a longitudinal relation.

### Methods

Relapse-onset MS patients with baseline CSF and spinal MRI available were retrospectively identified. Cross-sectional associations between intrathecal immunoglobulin production and cord lesions were assessed. For the subgroup where follow-up spinal MRIs were available, Andersen–Gill time-to-event models were used to estimate the risk of new cord lesions based on CSF makers and important covariates.

### Results

394 patients had baseline CSF and cord imaging available, of which 253 had CSF and serum IgG data available, 139 IgM data and 138 both IgG and IgM data. At baseline, patients with intrathecal IgG (IgG IF+; 175 out of 253) and IgM production (IgM IF+; 44 out of 139) had more cord lesions (IgG IF+, mean 1.30 vs. 0.85 spinal cord lesions, $p < 0.01$; IgM IF+ mean 1.64 vs. 0.91 spinal cord lesions, $p < 0.05$). After correction for important covariates, only IgM IF+ (IgM IF+ $\beta = 0.53$, $p = 0.02$; IgG IF+ $\beta = 0.38$, $p = 0.13$) was a statistically significant positive predictor for more spinal cord lesions at baseline. For 66 patients follow-up cord imaging was available with a

**Data availability statement:** Deidentified data for replication of the analyses and creation of tables/figures of this study are available in the Open Science Foundation data repository, https://osf.io/dq2nb/. The code of the analysis pipeline is available on https://github.com/danielkreiter/spinalcord-csf.

**Funding:** The author(s) received no specific funding for this work.

**Competing interests:** I have read the journal's policy and the authors of this manuscript have the following competing interests: Niels Franken has nothing to disclose. Daniel Kreiter, Demmie Bouweriks, Stephanie Knippenberg, Oliver Gerlach, Raymond Hupperts received institutional research grants and a grant from Nationaal MS fonds not related to this research; RH received fees for lectures and advisory boards from Biogen, Merck and Genzyme-Sanofi.

median follow-up time of 2 years (IQR 1.1–4.2). Longitudinally, there was no statistically significant association of IgM and IgG IF+ with new future cord lesions.

## Conclusion

While intrathecal IgM production is cross-sectionally independently associated with spinal cord lesions, this study does not support a role for predicting new cord lesions at follow-up.

---

## Introduction

The radiological follow-up of the spinal cord in multiple sclerosis (MS) remains a subject of ongoing controversy. Generally, spinal cord MRI is not part of routine follow-up imaging to detect silent radiological disease activity, as spinal cord lesions are mostly symptomatic [1]. Consequently, the added value of routine cord imaging at follow-up for detecting silent radiological activity seems limited. In addition, it leads to higher costs and places a greater burden on scanning resources. However, there is still a considerable proportion of MS patients that develop asymptomatic spinal cord lesions. These asymptomatic cord lesions can even occur in absence of new or active lesions in the brain. Earlier studies show that this is the case for 10% − 25% of new cord lesions at follow-up (i.e., that the asymptomatic cord lesion is the only sign of disease activity) [2–4]. These lesions are therefore missed when spinal cord MRI is not part of routine follow-up after starting disease-modifying treatment (DMT). Consequently, these patients may potentially be undertreated. However, simply adding routine spinal cord imaging to the disease monitoring regimen for all MS patients would not be efficient. Therefore, ideally, subgroups prone to more spinal cord involvement can be identified based on imaging- and/or biomarkers, in order to select candidates that would benefit from routine spinal imaging.

Biomarkers in cerebrospinal fluid (CSF) are widely used in the diagnostic process for multiple sclerosis, in large part since oligoclonal immunoglobulin G bands (OCB) in CSF have become part of diagnostic criteria [5] to meet the criterion of dissemination in time. In addition to OCB, intrathecal immunoglobulin G (IgG) and M (IgM) are often quantitatively determined in clinical practice when CSF is acquired in the diagnostic workup. Also, more recently, CSF markers like neurofilament light chain (Nfl; cytoskeletal protein of neurons), glial fibrillary acidic protein (GFAP; cytoskeletal protein of astrocytes) and kappa free light chain (κ-FLC; antibody subunits secreted by B cells) are receiving increasing attention in research and entering clinical practice [6,7]. Recently, κ-FLC became part of the 2024 McDonald criteria and can, like oligoclonal bands, replace dissemination in time [8]. With regard to the spinal cord, elevated intrathecal IgM has been shown to be cross-sectionally associated with more spinal cord involvement independent of brain lesion load [9,10]. This raises the question of whether it is also predictive of future spinal cord disease activity, as markers of future spinal cord involvement are currently lacking.

In this retrospective observational study, we investigate whether intrathecal immunoglobulins from routinely acquired CSF at diagnosis can be used to predict a higher likelihood of new spinal cord lesions at follow-up.

## Methods

Data was retrospectively extracted from the electronic health records of patients visiting the outpatient clinic at our tertiary MS center (Academic MS Center Zuyd, Zuyderland Medical Center, Sittard-Geleen, The Netherlands) between August 2007 and July 2024. Data were accessed for this study from the 28th of September 2024. At inclusion, data were deidentified. Patients were eligible for inclusion when (i) diagnosed with relapse-onset MS, (ii) availability (images) of at least one spinal cord and brain MRI, (iii) presence of CSF data (at least OCB determined and quantitatively determined IgG) with a maximum interval of one year from the baseline spinal cord MRI. Only patients with relapse-onset disease were included, since the baseline time point can be more accurately established in this group, thereby enhancing the homogeneity of the study cohort. Patients were excluded if (methyl)prednisolone was administered up to 30 days before CSF acquisition via lumbar puncture, as this could have an influence on the relevant CSF markers [11,12]. All patients were DMT naïve at the time of CSF sampling. To be included in the longitudinal analyses, patients needed to have at least one additional spinal cord MRI during follow-up (at least covering the cervical cord; six or more months after the baseline spinal cord MRI) and CSF data needed to be complete (OCB and quantitative IgM and IgG results available).

Extracted data consisted of age, sex, disease duration, Expanded Disability Status Scale (EDSS) scores, DMTs used, CSF results, brain MRI (date, whether there was new disease activity, number of lesions [0, 1–2, 3–8, 9+]) and spinal cord MRI data. Spinal MRI outcome data consisted of date, coverage (cervical, thoracic [including the conus medullaris] or both), number of lesions and lesion location. All spinal cord MRIs were also assessed by one of the investigators with experience in rating spinal cord MRIs (DK) at the time the database was constructed and before extraction of clinical and CSF data. In case of conflicts with the original radiological report, consensus was reached together with another rater (OG). In rare cases where original reports were inaccessible, the MRI was evaluated by the investigators (DK).

Throughout the extended duration of this retrospective study, MRI systems were subject to technical upgrades (such as the implementation of 3T magnets), accompanied by corresponding updates to scanning protocols. In earlier protocols, a proton density (PD) sequence was typically acquired alongside T2-weighted imaging, whereas in later protocols this was often replaced by short-tau inversion recovery (STIR) and/or phase-sensitive inversion recovery (PSIR) sequences. For spinal cord MRIs to be included, at least a sagittal T2 had to be available and one other sequence suitable for showing MS lesions (e.g., STIR/PD/PSIR, T2/T2* in axial direction) [13]. For the spinal cord, while 3T may improve interobserver agreement [14], there is currently no conclusive evidence that 3T increases lesion detection relative to 1.5T. Consequently, scans obtained at either field strength were considered eligible for inclusion [15,16].

The study was approved by the institutional research ethics committee (METCZ20240091). Informed consent was waived because of the retrospective nature of the study and the use of anonymized clinical data.

### CSF analysis

CSF and serum concentrations of IgG, IgM, and albumin were measured nephelometrically. Quantitative intrathecal Ig synthesis was calculated using the CSF-to-serum quotients and the Reiber formula [17,18] to determine the IgG and IgM intrathecal fraction [IF] in % (>0% meaning that there is intrathecal synthesis of the isotype). This method accounts for the integrity of the blood–CSF barrier using the CSF-to-serum quotient of albumin. OCB were determined using isoelectric focusing.

### Statistical analysis

To compare the number of baseline spinal cord lesions between the groups with and without intrathecal IgM and IgG production, a Mann-Withney U test was performed. Additionally, to correct for confounders (age, sex, disease duration,

number of brain lesions), a count regression model was applied using a negative binomial distribution (given overdispersion of the data). To analyze the relation with cerebral lesions at baseline an ordinal regression model was used due to the categorical nature of the brain MRI lesion number data available.

In the longitudinal analysis, Andersen–Gill time-to-event models for recurrent event data were applied to estimate the hazard ratio of new spinal lesions and brain lesions stratified by intrathecal immunoglobulin production. Schoenfeld's residual plots were used to assess the proportional hazards assumption and the variance inflation factor to check for collinearity. In the case of repeated events (i.e., multiple new cord lesions during follow-up), all occurrences were included in the time-to-event analyses. Here, included covariates were age, sex, spinal cord and brain lesions at baseline. To adjust for DMT exposure, the highest category of DMT used during follow-up was also included as covariate (none, low-, intermediate- or high-efficacy). Additionally, a sensitivity analysis with first instead of highest treatment category was performed. Low-efficacy treatments consisted of interferons, glatiramer acetate, teriflunomide; intermediate efficacy of dimethyl-, diroximel fumarate and the S1P inhibitors; high-efficacy treatments of natalizumab, ocrelizumab, rituximab, mitoxantrone, cladribine, alemtuzumab and ofatumumab. All statistical analyses were performed in R version 4.4.2.

## Results

An overview of screening and inclusion of patients is available in Fig 1. 394 patients had a baseline spinal cord MRI and CSF available. 141 of these patients were excluded because they had only OCB data available. IgM data was available for 139 patients and IgG data for 253 patients. 138 patients had complete CSF data (i.e., OCB, IgM and IgG data) of whom 66 had a follow-up spinal cord MRI available (65% of spinal cord MRIs in this subset also included the complete thoracic cord). All CSF acquisitions in our cohort were performed for diagnostic purposes. This also means that no patients were receiving a DMT when CSF was collected. Median time between CSF acquisition and baseline spinal cord MRI was 18 days (IQR 4–38). Intrathecal IgM production was seen in 44 out of 139 (31.7%) patients and intrathecal IgG production in 175 out of 253 (69.2%) patients.

Table 1 shows the patient characteristics stratified by CSF profile (for patients with complete CSF data available). The few patients (n = 5) that were negative for oligoclonal bands had a shorter disease duration and were older. Only one patient had IgM intrathecal synthesis without any indication of intrathecal IgG production (i.e., IgG IF- and no oligoclonal [IgG] bands).

### Baseline analysis

Patients with intrathecal IgG and IgM production had more spinal cord lesions (IgG IF+, mean of 1.3 vs. 0.9 spinal cord lesions, p = 0.001; IgM IF+ mean of 1.64 vs. 0.91 spinal cord lesions, p = 0.047; for interactions see Fig 2). When corrected for covariates (age, sex, disease duration, brain lesions), only intrathecal production of IgM was a statistically significant positive predictor for more spinal cord lesions at baseline (IgM IF+ β = 0.53, p = 0.02; IgG IF+ β = 0.38, p = 0.13). The number of brain lesions at baseline had no statistically significant association with IgG (OR 1.3, CI95% 0.6–2.7, p = 0.55) or IgM (OR 0.8, CI95% 0.4–1.6, p = 0.46) intrathecal production.

### Longitudinal analysis

In the group of patients with available longitudinal data, 66 patients had a total of 118 follow-up spinal cord MRIs with a median follow-up time of 2 years (IQR 1.1–4.2). In this group 18 patients used no DMT, 9 a low-efficacy, 13 an intermediate-efficacy and 26 a high-efficacy DMT. A total of 25 new cord lesions occurred at follow-up. Table 2 shows descriptives on the number and rate of new cord lesions per CSF profile. Of the patients developing new cord lesions, these lesions mostly occurred in the first few years (62% in the first year, 81% within the first three years). Intrathecal synthesis of IgG (HR 0.8, CI95% 0.3–2.2, p = 0.66) or IgM (HR 1.2, CI95% 0.4–3.7, p = 0.69) at baseline was not associated with a significantly higher number of new cord (or brain lesions) during follow-up (see Fig 3). In the sensitivity

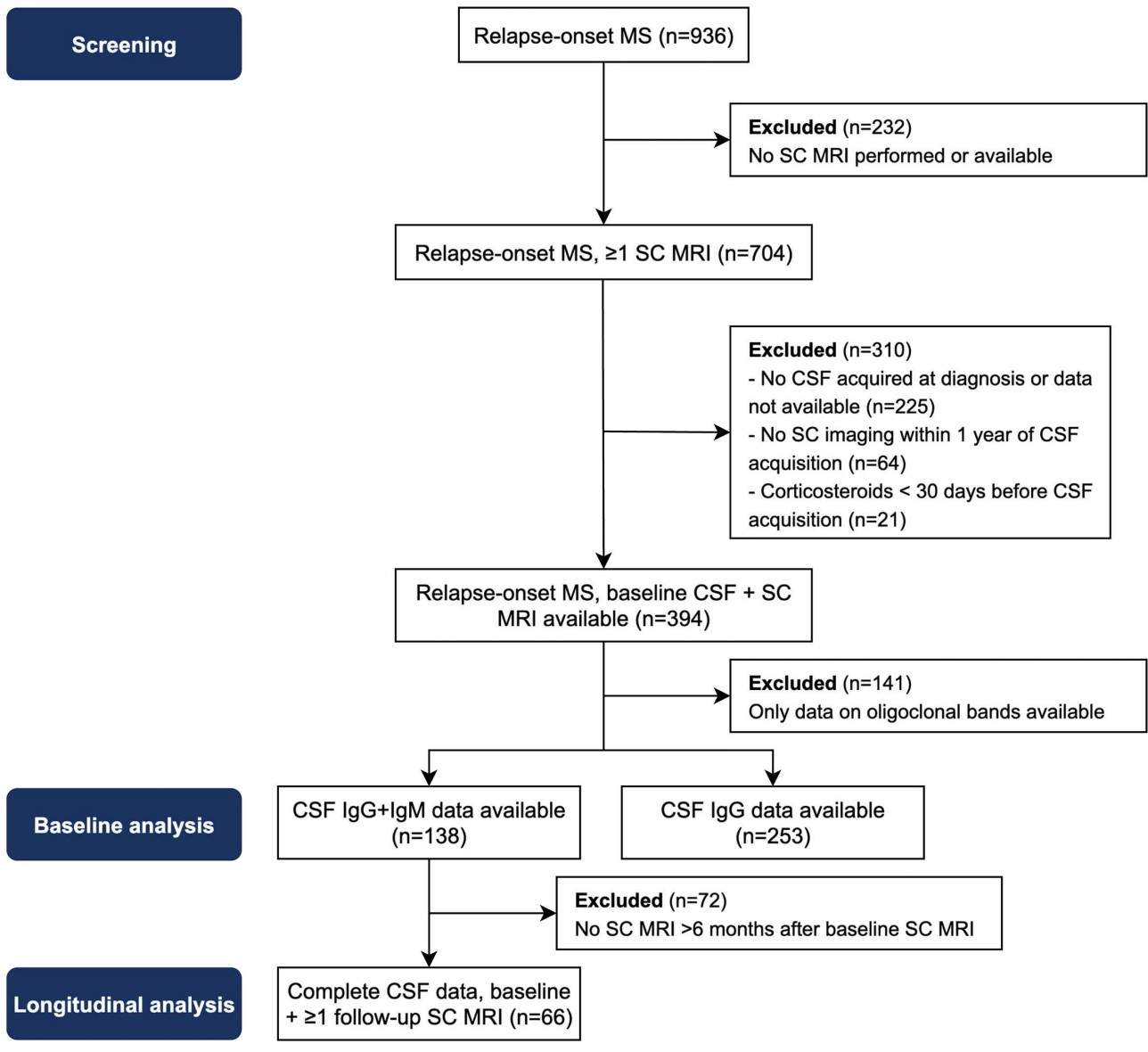

**Fig 1. Flow chart showing the screening of patients and selection for the baseline and longitudinal analyses.** SC = spinal cord, CSF = cerebrospinal fluid, IgG = immunoglobulin G, IgM = immunoglobulin M, MRI = magnetic resonance imaging.

analysis using the first treatment category used instead of the highest category, results were similar (HR 0.8, CI95% 0.3–2.3, p = 0.67 for IgG and HR 1.3, CI95% 0.5–3.1, p = 0.60 for IgM). From the covariates in the model, female sex (HR 5.55, CI95% 1.11–25.0, p < 0.05), younger age (HR 0.97 per year of age, CI95% 0.93–1.00, p < 0.05) and number of cord lesions at baseline (HR 1.41 per lesion, CI95% 1.00–1.97, p < 0.05) showed a possible relation with future cord lesions.

## Discussion

The present study confirms earlier findings that IgM intrathecal production is related to the presence of more spinal cord lesions at the time of CSF acquisition. This is in line with findings from Oechtering and collegeaus [9] who found a strong

**Table 1. Patient characteristics by CSF profile at baseline.**

| Characteristic | OCB-N = 5 | OCB + IgG-IgM- N = 22 | OCB + IgG-IgM+ N = 13 | OCB + IgG + IgM- N = 68 | OCB + IgG + IgM+ N = 30 |
|---|---|---|---|---|---|
| Age in years, median (Q1, Q3) | 40 (35, 41) | 41 (31, 46) | 35 (28, 45) | 34 (28, 45) | 31 (24, 38) |
| Disease duration in years, mean (SD) | 0.8 (1.0) | 2.6 (5.9) | 2.2 (4.8) | 1.7 (3.6) | 2.0 (3.1) |
| Sex, n (%) | | | | | |
| Female | 4 (80%) | 13 (59%) | 8 (62%) | 62 (91%) | 20 (67%) |
| Male | 1 (20%) | 9 (41%) | 5 (38%) | 6 (8.8%) | 10 (33%) |
| EDSS, median (Q1, Q3) | 1.8 (1.5, 2.0) | 1.5 (1.0, 2.3) | 1.5 (1.0, 2.0) | 1.5 (1.0, 2.0) | 2.0 (1.3, 3.0) |
| (Missing) | 3 | 6 | 4 | 17 | 6 |
| Number of spinal cord lesions, median (Q1, Q3) | 0.0 (0.0, 1.0) | 1.0 (0.0, 1.0) | 0.0 (0.0, 1.0) | 1.0 (0.0, 1.0) | 2.0 (0.0, 4.0) |
| Number of brain MRI lesions, n (%) | | | | | |
| 0 | 0 (0%) | 0 (0%) | 1 (7.7%) | 2 (2.9%) | 0 (0%) |
| 1-2 | 1 (20%) | 2 (9.1%) | 1 (7.7%) | 13 (19%) | 3 (10%) |
| 3-8 | 2 (40%) | 10 (45%) | 8 (62%) | 25 (37%) | 13 (43%) |
| 9+ | 2 (40%) | 10 (45%) | 3 (23%) | 28 (41%) | 14 (47%) |
| Relapses in year before baseline, median (Q1, Q3) | 1.0 (1.0, 1.0) | 1.0 (1.0, 1.0) | 1.0 (0.0, 1.0) | 1.0 (1.0, 1.0) | 1.0 (0.0, 1.0) |
| IgG index, median (Q1, Q3)* | 0.54 (0.51, 0.68) | 0.61 (0.54, 0.65) | 0.58 (0.56, 0.63) | 1.15 (0.87, 1.52) | 1.06 (0.83, 1.31) |
| IgM index, median (Q1, Q3)* | 0.05 (0.04, 0.14) | 0.08 (0.07, 0.12) | 0.58 (0.25, 0.78) | 0.09 (0.07, 0.12) | 0.51 (0.37, 0.71) |
| CSF leucocytes x $10^6$/L, median (Q1, Q3) | 3 (2, 15) | 5 (3, 8) | 3 (1, 6) | 10 (4, 15) | 10 (5, 19) |
| (Missing) | 0 | 1 | 1 | 0 | 0 |

EDSS = Expanded Disability Status Scale, OCB = oligoclonal bands. * for the IgG index a value of above 0.6 is often considered abnormal, and above 0.1 for the IgM index.

relation between intrathecal IgM production and a spinal cord syndrome as well as cord lesions on MRI. Similar to our study there was no association between IgG IF+ and cord lesions after correction for covariates. Additionally, no association was observed between brain lesions and intrathecal IgM or IgG production.

The main goal of this study was to examine whether intrathecal immunoglobulin production, particularly IgM, has longitudinal prognostic value and can potentially be used as a marker to identify patients who may need more frequent spinal cord imaging at follow-up. No evidence or trend was found suggesting a higher risk of new cord lesions at follow-up for the patients with IgM intrathecal synthesis at baseline. Also, in the descriptive data, the rate of new cord lesions was comparable for OCB + IgG + IgM- and OCB + IgG + IgM + CSF profiles and also when comparing OCB + IgG-IgM- and OCB + IgG-IgM + . Although not the focus of the current study, we found that among the covariates analyzed, being female, the number of cord lesions at baseline, and younger age were associated with the occurrence of new cord lesions at follow-up.

Over time various studies investigated the prognostic value of CSF markers including routinely performed assays like oligoclonal bands, quantitative IgM and IgG. More recently, research expanded towards markers like neurofilament light chain (Nfl), glial fibrillary acidic protein (GFAP) and kappa free light chain ($\kappa$-FLC). Studies that investigated the association of the routine set of CSF markers and future disease activity have shown mixed results. The outcomes of the literature summarised below mostly include time to disability milestones, time to next relapse or new brain MRI lesions. The longitudinal relation of these CSF markers specifically to future cord lesions has not been studied before. Quantitative intrathecal IgG has demonstrated predictive value for disability accrual/milestones in some studies [19], but no or only limited value in other studies [20,21]. For oligoclonal IgG bands again some studies show a shorter time to disability milestones [21,22], while others find no important difference in disability accrual [23]. Also, since OCBs are present in approximately 85% − 98% of patients [5,20,21], it is only relevant as a prognostic biomarker in the minority of MS patients where they are absent, possibly implying a milder disease course [20]. In regard to cerebral lesions on MRI, results have

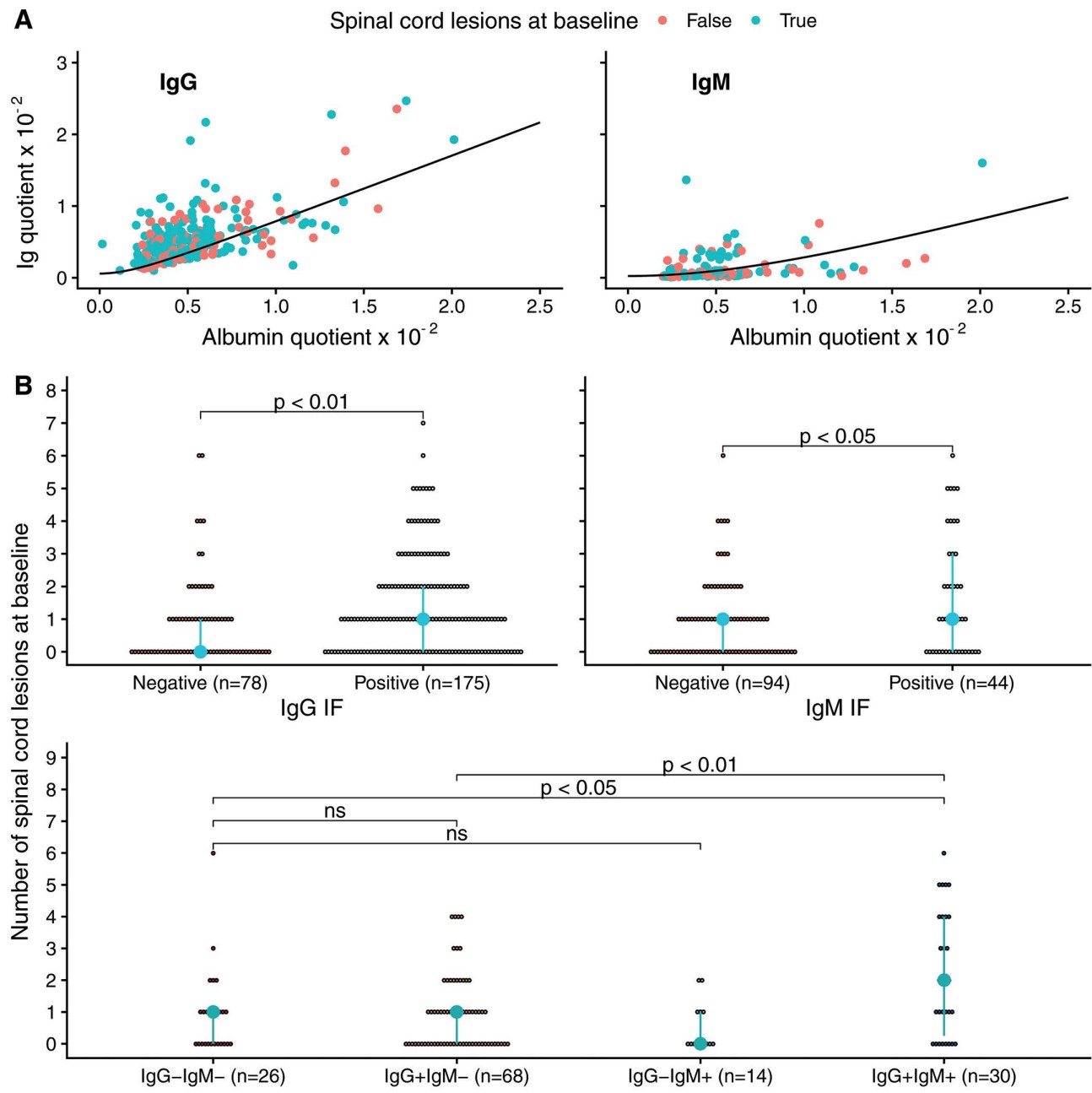

**Fig 2. Baseline associations between intrathecal immunoglobulin synthesis & spinal cord lesions.** (A) Reiber plots for IgG and IgM. Each dot represents a patient, with the red dots representing patients with no baseline lesions and the blue dots when there were cord lesions at baseline. The Ig quotient is the concentration of the immunoglobulin subtype in CSF divided by the concentration in serum. The albumin quotient the ratio between albumin in CSF and serum. The black line represents the threshold value above which there is intrathecal synthesis of the immunoglobulin subtype. (B) Presence of baseline spinal cord lesions in the groups with and without immunoglobulin intrathecal production. IF = intrathecal fraction. ns = not statistically significant.

also been mixed [20,24,25]. Intrathecal IgM synthesis is associated with higher disease activity (more new/enlarging lesions on brain MRI) and faster disability accrual [20,26–28]. On the other hand, a large study by Gasperi and colleagues found no association with EDSS disability milestones [19]. Nfl correlates well with disability across studies [7], but as to

**Table 2. New cord lesions at follow-up by CSF profile.**

| CSF profile | n | Number of new cord lesions | Total follow-up in years | New cord lesions per patient year |
|---|---|---|---|---|
| OCB- | 2 | 1 | 14.4 | 0.07 |
| OCB+IgG-IgM- | 11 | 3 | 39.5 | 0.08 |
| OCB+IgG-IgM+ | 7 | 1 | 14.9 | 0.07 |
| OCB+IgG+IgM- | 32 | 13 | 88.8 | 0.15 |
| OCB+IgG+IgM+ | 14 | 7 | 38.0 | 0.18 |

n = number of patients per CSF profile with longitudinal spinal cord MRI follow-up available.

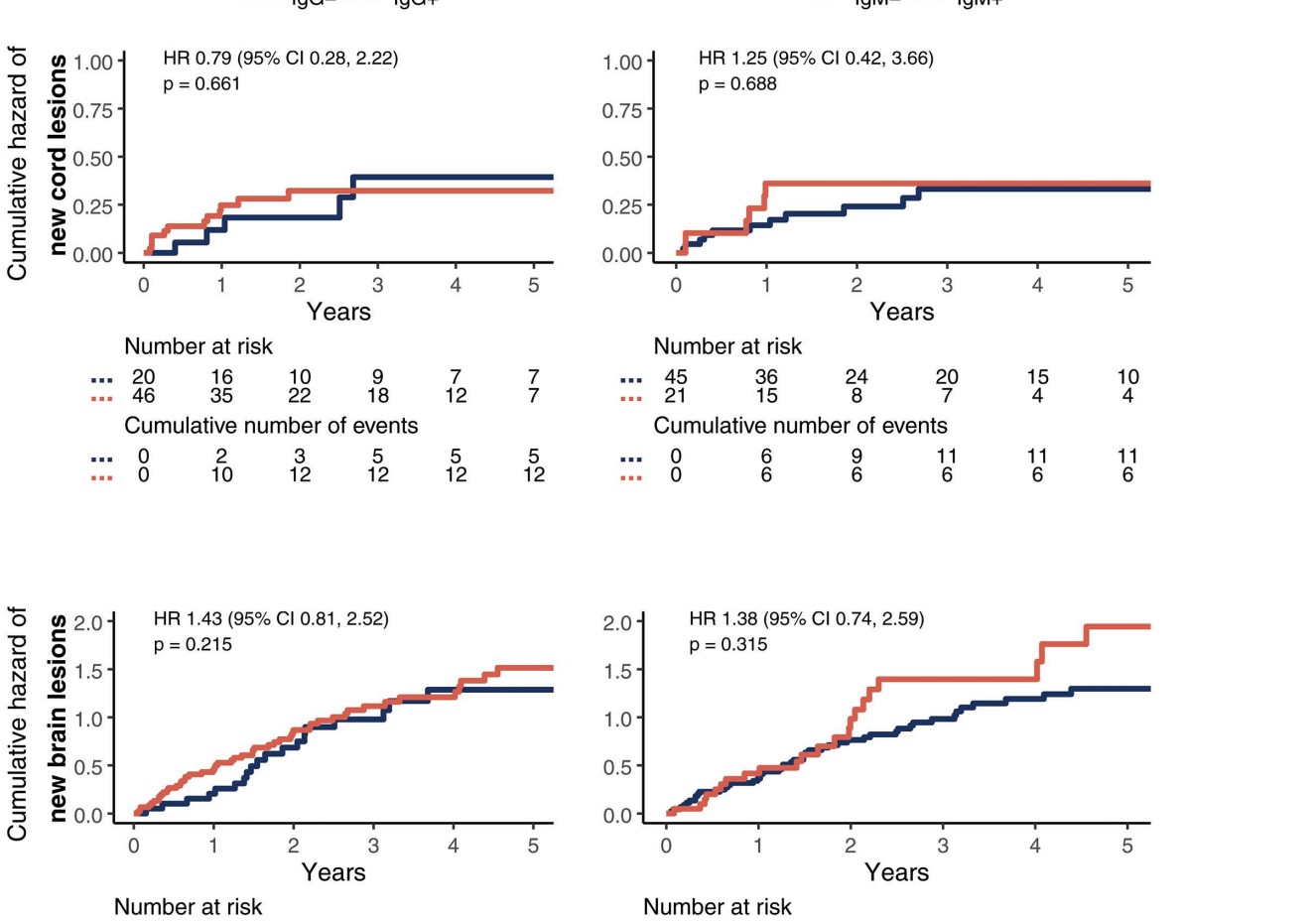

**Fig 3. Cumulative hazard plots and risk tables for spinal cord lesions (upper row) and brain lesions (lower row).** Patients with (red line) and without (blue line) intrathecal IgG production (left column) and IgM production (right column).

its prognostic value results have been mixed [7,29–34]. The same applies to GFAP [7,30,34]. But more recent and larger studies have made a good case that Nfl and GFAP (if appropriately standardized/corrected) may help predict the risk of disability progression [32,34]. $\kappa$-FLC indices have been shown to be a good predictor of time to a second clinical attack/ short-term disease progression [29,35–37], but there is no evidence yet showing longer term prognostic value [38]. It will be interesting to see in future studies if maybe Nfl, GFAP and/or $\kappa$-FLC can play a role in stratifying patients at risk of more cord involvement.

In summary, whether intrathecal immunoglobulin production has prognostic value remains ambiguous with mixed findings in the literature for clinical and brain MRI outcomes. When examining the cross-sectional relationship of intrathecal immunoglobulin production with disability and disease activity (as assessed through brain and/or spinal cord MRI), the existing evidence is more compelling [9,10,20,39–42]. Therefore, it can be argued that increased intrathecal immunoglobulins tell us more about the present (and maybe past) than about the future and that the detected intrathecal immunoglobulins are a response product to neuronal damage, but is not a driving actor in the complex immunological process leading to demyelination and neuroinflammation [43,44]. This is supported by studies showing that disrupted myelin binds immunoglobulins [45] and that the B-cell response is at least partly directed against intracellular autoantigens released during tissue destruction [43]. This could explain why a cross-sectional relation of lesion load with oligoclonal bands/intrathecal synthesis is more consistently shown, but that a longitudinal relationship is not found, as is the case in the present study.

## Limitations

The retrospective design of this study comes with limitations. A key limitation is incomplete availability of follow-up spinal cord MRI, reflecting local practice patterns rather than participant dropout. This reduces power and may introduce selection bias if imaging availability is associated with disease severity or symptoms. Although we adjusted for measured covariates, residual bias from informative missingness cannot be totally excluded. Over the years MRI sequences and scanners improved. Potentially this could lead to a spinal cord lesion on a follow-up scan to be considered new, while in fact it already existed but wasn't detected due to the possible poorer image quality of earlier scans. Nevertheless, this would have led to an overestimation in both groups and therefore should not have biased the results. Finally, while the use of different efficacy classes of DMTs was added as covariate to the models, DMT exposure was not randomized and our study was retrospective. Therefore, residual confounding, may persist despite covariate adjustment. These limitations should be considered when interpreting treatment-adjusted associations.

To address these limitations, currently, a prospective observational cohort study is ongoing with routine spinal cord imaging to identify early markers predictive for cord lesions at follow-up to help inform follow-up imaging strategies (https://clinicaltrials.gov/study/NCT06827834).

## Conclusion

In conclusion, while there is a cross-sectional relation between IgM intrathecal synthesis and cord lesions, this study does not support that intrathecal immunoglobulin production can be used to stratify patients that need more routine follow-up imaging of the spinal cord.

## Author contributions

**Conceptualization:** Daniel Kreiter, Niels Franken, Stephanie Knippenberg, Raymond Hupperts, Oliver Gerlach.

**Data curation:** Daniel Kreiter.

**Formal analysis:** Daniel Kreiter, Niels Franken.

**Methodology:** Daniel Kreiter, Niels Franken.

**Supervision:** Oliver Gerlach.

**Writing – original draft:** Daniel Kreiter, Niels Franken.

**Writing – review & editing:** Demmie Bouweriks, Stephanie Knippenberg, Raymond Hupperts, Oliver Gerlach.

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
