## [Decision Letter · Decision Letter 0]

18 Sep 2025

Dear Dr. Kreiter,

We look forward to receiving your revised manuscript.

Kind regards,

Karlo Toljan

Academic Editor

PLOS ONE

Journal Requirements:

“I have read the journal's policy and the authors of this manuscript have the following competing interests: Niels Franken has nothing to disclose. Daniel Kreiter, Demmie Bouweriks, Stephanie Knippenberg, Oliver Gerlach, Raymond Hupperts received institutional research grants and a grant from Nationaal MS fonds not related to this research; RH received fees for lectures and advisory boards from Biogen, Merck and Genzyme-Sanofi.”

We note that you received funding from a commercial source: Nationaal MS fonds

5. Please include captions for your Supporting Information files at the end of your manuscript, and update any in-text citations to match accordingly. Please see our Supporting Information guidelines for more information: http://journals.plos.org/plosone/s/supporting-information .

Additional Editor Comments:

Reviewer's Responses to Questions

**Comments to the Author**

1. Is the manuscript technically sound, and do the data support the conclusions?

Reviewer #1: Yes

Reviewer #2: Partly

2. Has the statistical analysis been performed appropriately and rigorously?

Reviewer #1: Yes

Reviewer #2: Yes

3. Have the authors made all data underlying the findings in their manuscript fully available?

Reviewer #1: Yes

Reviewer #2: Yes

4. Is the manuscript presented in an intelligible fashion and written in standard English?

Reviewer #1: Yes

Reviewer #2: Yes

Reviewer #1: In the manuscript "Spinal cord lesions in relapsing-remitting multiple sclerosis: prognostic value of

intrathecal production of immunoglobulins", Kreiter et al, aimed to evaluate the prognostic value of CSF biomarkers predicting new spinal cord lesions on follow up imaging in MS. This is an important and interesting topic which requires exploration. Although this study has several limitations (notably the retrospective nature of the study) the results are valuable to scientific community to prompt further evaluation once the following recommendations to improve the quality of the manuscript are addressed as listed below:

Major:

- Methods: Please provide further explanation why progressive MS patients were excluded.

- Methods: Please include MRI protocols including sequences obtained, magnet strength, etc.

- Methods: Please include normal cut-off values for CSF IgM and IgG indices.

- Methods: I don't believe impact of DMT is adequately addressed in the current study. Evaluating the first DMT used (vs. highest efficacy DMT used) would make more sense for the current study. Further, would it be possible to compare impact of DMT efficacy using propensity score matching?

- Methods: Would it be possible to provide data on lesion volumes in addition to lesion counts?

- Methods: It remains unclear exactly how recurrent events were analyzed in this model. Were adjustments such Andersen–Gill model used or some other methods? Were there any violations in the assumptions of the CPH model?

- Discussion: Impact of DMT on IgG and IgM index warrants further discussion. Could this impact the prognostic value of these biomarkers if they are considered dynamic and DMT dependent?

Minor:

- Abstract: remove “more” in the line “… identify patients prone to more future …”

- Abstract: prefer inclusion of number included participants and baseline demographics if possible within word/character count limits

Reviewer #2: In this study, the authors investigated the association between intrathecal IgM production and both cross-sectional and longitudinal changes in spinal cord lesions in patients with relapsing-onset multiple sclerosis (MS). They report that patients with intrathecal IgG and IgM production had more spinal cord lesions compared to those without. However, in a subgroup of 66 patients with longitudinal spinal cord MRI follow-up (118 scans, median follow-up ~2 years), no association was found between intrathecal IgM and the development of new spinal cord lesions.

While the study addresses a relevant question, several concerns remain:

• In the longitudinal evaluation, it is important to specify which spinal cord regions were covered by MRI. Was there variability in spinal cord coverage across patients?

• The potential use of heterogeneous spinal cord MRI protocols across follow-up scans might be a relevant confounding factor.

• As acknowledged by the authors, disease-modifying therapies (DMTs) represent a potential confounder that is not sufficiently addressed in the analyses.

• The retrospective derivation of spinal cord lesion burden from radiology reports is a major limitation. In how many cases were MRI images re-evaluated by the authors?

• Given the distribution of lesion count, multivariate linear regression may not be the most appropriate modeling approach.

• Cox proportional hazards models estimate hazard ratios, not risk ratios as stated in the manuscript, and this terminology should be corrected.

• The statement “66 patients had a total of 118 follow-up spinal cord MRIs” corresponds to fewer than two MRIs per patient on average, which seems low and requires clarification.

• While a relatively large number of participants contributed to the cross-sectional analysis, a major limitation of the study is the substantial drop-out in the longitudinal analyses.

Minor:

• In the abstract, correct “IgF” to “IgG.”

• The abstract should indicate the total number of patients included, as well as details on follow-up duration.

**Do you want your identity to be public for this peer review?** For information about this choice, including consent withdrawal, please see our Privacy Policy

Reviewer #1: No

Reviewer #2: No

---

## [Author Response · Author response to Decision Letter 1]

20 Oct 2025

Reviewer #1: Please provide further explanation why progressive MS patients were excluded.

Authors: Only relapse-onset patients and not progressive-onset were selected as the baseline moment can be better determined for relapse-onset patient and to achieve a more homogeneous group. The explanation has been added to the Methods section as well (lines 86-88).

Reviewer #1: Please include MRI protocols including sequences obtained, magnet strength, etc.

Authors: In the Methods-section we added a section giving an explanation on the imaging protocols and scanners. Additionally, a section to the Limitations-section was added to explain the limitation this causes. (lines 105-115)

Reviewer #1: Please include normal cut-off values for CSF IgM and IgG indices.

Authors: The relevant cut-off value used is already mentioned in the Methods-section, when using the Reiber formula, the cut-off is above 0% to consider the value of the CSF IgG and IgM as high correcting for blood-brain barrier permeability and serum IgG and IgM. For a detailed description of this method see Reiber, 1998. We noticed the reference to the Reiber papers to be missing in the methods section, these have now been added. The mentioned cut-off is the cut-off used in other work as well (e.g. Oechtering 2021 and 2022). Nevertheless, we expanded the explanation a bit, to give a global idea of what the goal of using the Reiber formula is (lines 122-126).

In the baseline characteristics table (Table 1) we now added often used cut-offs for the mentioned IgG and IgM index in the table.

Reviewer #1: I don't believe impact of DMT is adequately addressed in the current study. Evaluating the first DMT used (vs. highest efficacy DMT used) would make more sense for the current study. Further, would it be possible to compare impact of DMT efficacy using propensity score matching?

Authors: With the reviewers suggestion we decided to add a sensitivity analysis using first instead of highest treatment category. This did not result in a relevant difference in the results. We added this to the main text, see linesn 208-210. In regard to the comment whether it would be possible to compare impact of DMT categories on cord lesions using propensity score matching, that is not the goal of this study as we are not comparing interventions but looking into a baseline predictor (for the DMT effect question analyzed on this database using propensity score matching, see Kreiter et al, 2023).

Reviewer #1: Would it be possible to provide data on lesion volumes in addition to lesion counts?

Authors: This would certainly be interesting, but unfortunately this is not possible from current data, as the performed clinical spinal cord MRI protocols did not include 3D sequences to be able to determine lesion volumes.

Reviewer #1: Impact of DMT on IgG and IgM index warrants further discussion. Could this impact the prognostic value of these biomarkers if they are considered dynamic and DMT dependent?

IgG and IgM indices were determined using CSF at diagnosis (so before start of treatment), which means there is no impact of DMTs there (this is already mentioned in the manuscript, lines 158-159). However, in some cases patients could have been treated with corticosteroids shortly before CSF acquisition, which can impact these indices. Therefore, these patients were excluded (n=21, see figure 1 in the manuscript).

Reviewer #1: Methods: It remains unclear exactly how recurrent events were analyzed in this model. Were adjustments such Andersen–Gill model used or some other methods? Were there any violations in the assumptions of the CPH model?

Authors: The reviewer is right that in fact the model applied here is an Andersen-Gill model, as the analysis uses coxph(Surv(start, stop, event) ~ X1 + …, id=PATIENT_ID, data = data) command in R to allow multiple events per subject and account for within-subject correlation. We corrected the terminology used in the Methods-section (lines 135-137).

There were no violations of the assumptions of the model. Schoenfeld’s residual plots were used to assess the proportional hazards assumption and the variance inflation factor to check for collinearity. A mention that this has been checked has now been added to the manuscript (lines 137-138).

Reviewer #1: Abstract: remove “more” in the line “… identify patients prone to more future …”

Authors: Thank you for the suggestion, this has been removed from that sentence.

Reviewer #1: Abstract: prefer inclusion of number included participants and baseline demographics if possible within word/character count limits

Authors: We added a sentence with inclusion numbers to the Results-part of the Abstract.

Reviewer #2: In the longitudinal evaluation, it is important to specify which spinal cord regions were covered by MRI. Was there variability in spinal cord coverage across patients?

Authors: Yes there was variation, the thoracic cord was not always included, and this is mentioned in the manuscript, “65% of spinal cord MRIs in this subset also included the complete thoracic cord” (lines 157-158).

Reviewer #2: The potential use of heterogeneous spinal cord MRI protocols across follow-up scans might be a relevant confounding factor.

This is true and this is now more extensively discussed in the Limitations-section (lines …)

Reviewer #2: The retrospective derivation of spinal cord lesion burden from radiology reports is a major limitation. In how many cases were MRI images re-evaluated by the authors?

Authors: All spinal cord MRI reports and images were checked by one of the investigators with experience with spinal cord imaging (DK), when the database was constructed (also before adding clinical / CSF data). In case of conflicts with the original report, a third rater was consulted (OG). This has now been better explained within the text of the manuscript (lines 282-286).

Reviewer #2: Given the distribution of lesion count, multivariate linear regression may not be the most appropriate modeling approach.

Authors: As lesion count is the dependent variable of the model here, the distribution of this variable is not of relevance to the assumptions of the regression model.

Reviewer #2: Cox proportional hazards models estimate hazard ratios, not risk ratios as stated in the manuscript, and this terminology should be corrected.

Authors: Thank you, this is correct and has now been corrected in the text.

Reviewer #2: The statement “66 patients had a total of 118 follow-up spinal cord MRIs” corresponds to fewer than two MRIs per patient on average, which seems low and requires clarification.

Authors: It concerns 118 follow-up MRIs (so in addition to the baseline cord MRI), this is a bit fewer than 2 follow-up cord MRIs per patient, but given the median follow-up time of 2 years this is not that low.

While a relatively large number of participants contributed to the cross-sectional analysis, a major limitation of the study is the substantial drop-out in the longitudinal analyses.

Reviewer #2: In the abstract, correct “IgF” to “IgG.”

Authors: This has been corrected in the new version of the manuscript.

Reviewer #2: The abstract should indicate the total number of patients included, as well as details on follow-up duration.

Authors: This is now added to the abstract (lines 29-30 & 35-36).

---

## [Decision Letter · Decision Letter 1]

4 Nov 2025

Dear Dr. Kreiter,

Thank you for submitting your manuscript to PLOS ONE. After careful consideration, we feel that it has merit but does not fully meet PLOS ONE’s publication criteria as it currently stands. Therefore, we invite you to submit a revised version of the manuscript that addresses the points raised during the review process.

We look forward to receiving your revised manuscript.

Kind regards,

Karlo Toljan

Academic Editor

PLOS ONE

Journal Requirements:

Reviewers' comments:

Reviewer's Responses to Questions

**Comments to the Author**

Reviewer #1: All comments have been addressed

Reviewer #2: (No Response)

2. Is the manuscript technically sound, and do the data support the conclusions?

Reviewer #1: Yes

Reviewer #2: Yes

3. Has the statistical analysis been performed appropriately and rigorously?

Reviewer #1: Yes

Reviewer #2: No

4. Have the authors made all data underlying the findings in their manuscript fully available?

Reviewer #1: Yes

Reviewer #2: Yes

5. Is the manuscript presented in an intelligible fashion and written in standard English?

Reviewer #1: Yes

Reviewer #2: Yes

Reviewer #1: Thank you for addressing the comments. A few additional minor suggestions outlined below.

I would suggest removing the following sentence:

“With respect to the spinal cord, there is no conclusive evidence of a difference in lesion detection between 1.5T and 3T MRI systems.”

Please present number of scans at each magnetic field instead. Please see manuscript: "Impact of 3 Tesla MRI on interobserver agreement in clinically isolated syndrome: A MAGNIMS multicentre study."

Please add a sentence in the methods to specify that patients were DMT-naïve at the time of CSF testing.

Table 1: There is mention of IgM twice. The first should be IgG.

Reviewer #2: • It seems that the authors did not address the point: “As acknowledged by the authors, disease-modifying therapies (DMTs) represent a potential confounder that is not sufficiently addressed in the analyses.”

• In response to the comment “Given the distribution of lesion count, multivariate linear regression may not be the most appropriate modeling approach,” the authors state that “As lesion count is the dependent variable of the model here, the distribution of this variable is not of relevance to the assumptions of the regression model.” This statement appears inaccurate, as the distribution of the dependent variable is crucial for ensuring that model assumptions are met. Depending on its distribution, a count regression model (e.g. Poisson or negative binomial) may be more appropriate.

• According to the authors’ response, the point “While a relatively large number of participants contributed to the cross-sectional analysis, a major limitation of the study is the substantial drop-out in the longitudinal analyses” also seems not to have been addressed.

**Do you want your identity to be public for this peer review?** For information about this choice, including consent withdrawal, please see our Privacy Policy

Reviewer #1: No

Reviewer #2: No

---

## [Author Response · Author response to Decision Letter 2]

7 Nov 2025

Dear editor,

We would like to thank the reviewers for their replies and additional comments. The comments are addressed below:

Review #1: I would suggest removing the following sentence:

“With respect to the spinal cord, there is no conclusive evidence of a difference in lesion detection between 1.5T and 3T MRI systems.”

Please present number of scans at each magnetic field instead. Please see manuscript: "Impact of 3 Tesla MRI on interobserver agreement in clinically isolated syndrome: A MAGNIMS multicentre study."

Authors: We appreciate the reviewer’s suggestion and highlighting the MAGNIMS multicentre study. We agree that 3T systems can improve signal-to-noise and, as reported in the mentioned paper, may yield higher interobserver agreement for spinal cord lesions. However, that study also notes that it has not conclusively been demonstrated that 3T achieves higher lesion detection than lower field strengths in spinal cord MRI. Our sentence was intentionally phrased to reflect this nuance.

Reporting the number of scans acquired at 1.5T versus 3T in our dataset would not address the reviewer’s core concern. Field-strength counts primarily reflect practice patterns and site-level availability, not diagnostic yield. Our study was neither designed nor powered to perform a head-to-head test of lesion detection across field strengths, and presenting raw counts could be misinterpreted as evidence for, or against, superiority.

For these reasons, we respectfully propose to retain the sentence. To acknowledge the reviewer’s point and improve clarity, we revised the surrounding text to make the distinction explicit and added a reference to the study mentioned (lines 113-116 in clean version): “For the spinal cord, while 3T may improve interobserver agreement [14], there is currently no conclusive evidence that 3T increases lesion detection relative to 1.5T. Consequently, scans obtained at either field strength were considered eligible for inclusion [15, 16].”

Review #1: Please add a sentence in the methods to specify that patients were DMT-naïve at the time of CSF testing.

Authors: We thank the reviewer for this suggestion. We have added the following sentence to the Methods (lines 90-91):

“All patients were disease-modifying therapy (DMT)–naïve at the time of CSF sampling.”

Review #1: Table 1: There is mention of IgM twice. The first should be IgG.

Authors: We appreciate the careful reading. This has been corrected from IgM to IgG in Table 1. We have reviewed the table, caption, and corresponding in-text references to ensure consistency throughout.

Reviewer #2: It seems that the authors did not address the point: “As acknowledged by the authors, disease-modifying therapies (DMTs) represent a potential confounder that is not sufficiently addressed in the analyses.”

Authors: We thank the reviewer for raising this important issue and apologize for not addressing it adequately in our previous reply. For the longitudinal analyses we adjusted for DMT exposure by including DMT category as a covariate in the time-to-event models (we clarified the explanation a bit in lines 141-143 in the clean version). Given the available data, sample size, and the retrospective study design, we believe this represents the most robust adjustment feasible with our dataset. We fully agree, however, that residual confounding cannot be completely eliminated in nonrandomized, retrospective data. We therefore explicitly acknowledge this limitation in the revised text in the limitations section (lines 286 – 290 in the clean version). Importantly, this concern does not affect the baseline analyses, as all patients were DMT-naïve at baseline.

We hope these clarifications address the reviewer’s concern while accurately reflecting the strengths and constraints of our dataset.

Reviewer #2: In response to the comment “Given the distribution of lesion count, multivariate linear regression may not be the most appropriate modeling approach,” the authors state that “As lesion count is the dependent variable of the model here, the distribution of this variable is not of relevance to the assumptions of the regression model.” This statement appears inaccurate, as the distribution of the dependent variable is crucial for ensuring that model assumptions are met. Depending on its distribution, a count regression model (e.g. Poisson or negative binomial) may be more appropriate.

Authors: We thank the reviewer for this important correction and apologize for our imprecise statement. Accordingly, we have re-estimated the lesion-count models using count regression. Because the data were overdispersed, we employed a negative binomial model (using glm.nb in R) rather than Poisson. The Methods section has been amended to describe the modelling approach (lines 130-133 in the clean version), and the Results have been updated accordingly (lines 180-181 in the clean version). These changes produced minor numerical differences but did not alter the substantive conclusions of the analysis. We appreciate the reviewer’s guidance, which has improved the statistical rigor and clarity of the manuscript.

Reviewer #2: According to the authors’ response, the point “While a relatively large number of participants contributed to the cross-sectional analysis, a major limitation of the study is the substantial drop-out in the longitudinal analyses” also seems not to have been addressed.

Authors: We thank the reviewer for highlighting this point and apologize for not addressing it sufficiently in our earlier reply. To clarify, the issue is not loss to clinical follow-up, but lack of follow-up spinal cord MRI. Serial brain MRI is routine, while spinal cord MRI is not. Therefore, some participants had no follow-up cord imaging despite ongoing clinical follow-up. This expected pattern explains the smaller longitudinal imaging cohort. These limitations are now explicitly discussed in the Limitations section (lines 278-282 in the clean version), where we have expanded the explanation to address the reviewer’s point,

---

## [Editor Report · Decision Letter 2]

11 Nov 2025

Spinal cord lesions in relapsing-remitting multiple sclerosis: prognostic value of intrathecal production of immunoglobulins

PONE-D-25-41819R2

Dear Dr. Kreiter,

We’re pleased to inform you that your manuscript has been judged scientifically suitable for publication and will be formally accepted for publication once it meets all outstanding technical requirements.

Kind regards,

Karlo Toljan

Academic Editor

PLOS ONE
---

## [Editor Report · Acceptance letter]

PONE-D-25-41819R2

PLOS ONE

Dear Dr. Kreiter,

I'm pleased to inform you that your manuscript has been deemed suitable for publication in PLOS ONE. Congratulations! Your manuscript is now being handed over to our production team.

Kind regards,

on behalf of

Dr. Karlo Toljan

Academic Editor

PLOS ONE